# Investigation of Advanced Glycation End-Products, α-Dicarbonyl Compounds, and Their Correlations with Chemical Composition and Salt Levels in Commercial Fish Products

**DOI:** 10.3390/foods12234324

**Published:** 2023-11-29

**Authors:** Lihong Niu, Shanshan Kong, Fuyu Chu, Yiqun Huang, Keqiang Lai

**Affiliations:** 1School of Food Engineering, Ludong University, No. 186 Middle Hongqi Road, Yantai 264025, China; 2College of Food Science and Technology, Shanghai Ocean University, No. 999 Hucheng Huan Road, Lingang New City, Shanghai 201306, China; 18703730812@163.com (S.K.); c17864755417@163.com (F.C.); 3Engineering Research Center of Food Thermal-Processing Technology, Shanghai Ocean University, Shanghai 201306, China; 4School of Food Science and Bioengineering, Changsha University of Science & Technology, 960, 2nd Section, Wanjiali South Road, Changsha 410114, China; yiqunh@csust.edu.cn

**Keywords:** N^ε^-carboxymethyllysine, N^ε^-carboxyethyllysine, glyoxal, methylglyoxal, salt, fish product, correlation analysis

## Abstract

The contents of free and protein-bound advanced glycation end-products (AGEs) including N^ε^-carboxymethyllysine (CML) and N^ε^-carboxyethyllysine (CEL), along with glyoxal (GO), methylglyoxal (MGO), chemical components, and salt in commercially prepared and prefabricated fish products were analyzed. Snack food classified as commercially prepared products exhibited higher levels of GO (25.00 ± 3.34–137.12 ± 25.87 mg/kg of dry matter) and MGO (11.47 ± 1.39–43.23 ± 7.91 mg/kg of dry matter). Variations in the contents of free CML and CEL increased 29.9- and 73.0-fold, respectively. Protein-bound CML and CEL in commercially prepared samples were higher than those in raw prefabricated ones due to the impact of heat treatment. Levels of GO and MGO demonstrated negative correlations with fat (R = −0.720 and −0.751, *p* < 0.05) in commercially prepared samples, whereas positive correlations were observed (R = 0.526 and 0.521, *p* < 0.05) in raw prefabricated ones. The heat-induced formation of protein-bound CML and CEL showed a negative correlation with the variations of GO and MGO but was positively related to protein levels in prefabricated products, suggesting that GO and MGO may interact with proteins to generate AGEs during heating. The influence of NaCl on the formation of GO and MGO exhibited variations across different fish products, necessitating further investigation.

## 1. Introduction

Endogenous food contaminants are naturally formed through biochemical reactions in raw ingredients during processing and storage, posing a significant risk to human health due to their inherent toxicity and harmful nature [1]. These contaminants have intricate compositions and concealed origins, and their accumulation within the human body can result in various adverse toxic effects. Seafood products with high protein contents are susceptible to non-enzymatic glycation reactions during processing and storage, leading to the formation of advanced glycation end-products (AGEs), wherein N^ε^-carboxymethyllysine (CML) and N^ε^-carboxyethyllysine (CEL) serve as two representative compounds [2,3,4]. Dietary AGEs, as a representative class of endogenous food contaminants, have garnered increasing attention in recent years. Research has demonstrated that, upon ingestion by the human body, dietary AGEs can form cross-links with proteins and an extracellular matrix within cells, thereby modifying their molecular structure and function [5]. Furthermore, AGEs are capable of inducing oxidative stress within cells, resulting in cellular damage and potentially contributing to the development of various chronic diseases such as diabetes and its complications, uremia, atherosclerosis, and Alzheimer’s disease [6]. Foods that are high in protein and fat, such as meat, eggs, and dairy products, typically contain significantly higher levels of AGEs compared to low-protein and low-fat foods like fruits, grains, and vegetables [7,8]. Furthermore, the levels of AGEs in food can also be influenced by various food processing methods and storage conditions [4,9]. 

The formation pathways of dietary AGEs are intricate, with the Maillard reaction pathway serving as a prototypical example. This pathway involves the condensation reaction between the unbound amino group of proteins and the carbonyl group of reducing sugars, resulting in the generation of Schiff bases [8]. These Schiff bases subsequently undergo Amadori rearrangement and subsequent oxidation to form AGEs. Alternatively, the initial and intermediate products of the Maillard reaction, specifically Schiff bases and Amadori rearrangement products, can undergo oxidation to form α-dicarbonyl compounds such as glyoxal (GO) and methylglyoxal (MGO). These compounds subsequently react with the free amino groups of proteins to generate AGEs [10,11]. Additionally, the auto-oxidation of reducing sugars and the peroxidation of lipids can also contribute to the generation of reactive α-carbonyl compounds to participate in the formation of AGEs [12]. Considering seafood as a complex food system, comprising various components such as proteins and lipids, the utilization of different types and dosages of additives during processing, along with variations in processing and storage conditions, may impact the pathways of AGE formation in seafood. The precise mechanisms underlying these effects pose challenges for a definitive explanation.

The protein and lipid oxidation reactions that occur during the processing and storage of seafood, which is rich in proteins and polyunsaturated fatty acids, can result in the formation of elevated levels of AGEs, thereby raising certain safety concerns [4]. The production volume of seafood is substantial, and it is commonly available in the market in the form of cured and seasoned products. The levels of CML and CEL in 20 commercially available canned fish products were investigated by Zhao et al. [13], along with their correlation to protein and polyunsaturated fatty acid content. The findings revealed significant variations in the levels of CML and CEL among different canned fish products, with maximum differences reaching 131- and 13-fold, respectively. The content of proteins and specific polyunsaturated fatty acids exhibited a significant correlation with the levels of CML or CEL, suggesting that the composition of raw materials and exogenous additives can impact the formation of AGEs in seafood products. Sodium chloride (NaCl) is the most commonly used ingredient during the processing of seafood products. Studies have shown that NaCl can induce protein and lipid oxidation in muscle food systems [14]; however, it can also act as an antioxidant depending on its concentration, pH, and other factors [15]. The improving effect of NaCl on AGE formation has been reported in pork [16] and beef [17] products, whereas other studies have demonstrated that salt had no significant impact on the levels of CML in boiled beef [18]. The underlying impact mechanism of NaCl on the formation of AGEs, particularly in relation to their precursors (e.g., GO and MGO), remains unclear and requires further investigation. 

This study investigated the levels of α-dicarbonyl compounds (GO and MGO) and free and protein-bound AGEs (CML and CEL) in commercially prepared and prefabricated fish products. Additionally, the study analyzed the contents of basic components in the fish products, particularly focusing on salt, and examined their correlations with the levels of GO, MGO, CML, and CEL. The objective was to explore the factors influencing AGE levels in seafood products while providing valuable insights for seafood processing and promoting healthy dietary choices.

## 2. Materials and Methods

### 2.1. Chemicals

The four standards (CML, CEL, D_4_-CML, and D_4_-CEL, all ≥ 98%) for AGE detection were obtained from Toronto Research Chemicals Inc. (Toronto, ON, Canada). Two α-dicarbonyl compound standards, namely, GO (40.9%) and MGO (40%), were purchased from Chem Service Inc. (Westchester, NY, USA) and Fluorochem Ltd. (Hadfield, UK), respectively. The internal standard 5-Methylquinoxaline (5-MQ) for GO and MGO detection was obtained from J&K Scientific LLC (Shanghai, China). All other chemicals were sourced from Sinopharm Chemical Reagent CO., Ltd. (Shanghai, China).

### 2.2. Sample Preparation 

A total of 32 different fish products, including 13 commercially prepared products and 19 prefabricated products, were collected from various aquatic product markets and supermarkets in Shanghai, China. Detailed information regarding these samples is available in Table 1. The commercially prepared food samples were directly ground and prepared for analysis. As for the prefabricated products, each sample was ground and sealed into a cylindrical aluminum cell (internal height: 5 mm, inner diameter: 50 mm), which was specially designed to facilitate rapid heat transfer and ensure even heat distribution [19]. The cells were then subjected to a 15 min heating process in boiling water at 100 °C. The heating temperature is commonly used for home cooking, and a cooking duration of 15 min is sufficient for the central temperature of the heated sample to approach approximately 100 °C [19]. Subsequently, the heated samples were ground for analysis. 

### 2.3. Chemical Composition Analysis

The moisture, protein, and fat contents of the collected fish products were determined following the AOAC method [3,20]. The salt content of each sample was measured based on the method described by Qiu et al. [21]. Approximately 2.00 g of the ground sample was mixed with 10 mL of water, thoroughly ground, and allowed to stand for 2 h. Afterward, the mixture was centrifuged at 2000× *g* for 10 min (Rotina 380R, Hettich company, Kirchlengern, Germany). The resulting supernatant was collected, and titration was performed using a 0.1 mol/L AgNO_3_ standard solution, with K_2_CrO_4_ used as an indicator. The chloride ion (Cl^−^) concentration was determined based on the volume of the AgNO_3_ solution consumed. Every gram of Cl^−^ is equal to 1.65 g of NaCl.

### 2.4. Extraction and Detection of GO and MGO

The extraction and detection of the two α-dicarbonyl compounds, GO and MGO, were conducted following the methods outlined by Li et al. [22] and Chu et al. [23]. Approximately 1.00 g of the ground sample was homogenized with 10 mL of 0.6 mol/L perchloric acid for 1 min, allowed to settle for 30 min at 4 °C, and then centrifuged for 10 min at 8000× *g* and 4 °C. Supernatant of 1 mL was mixed with 0.2 mL of a 5 mg/mL o-phenylenediamine solution and reacted for 3 h in the dark at a water bath set to 60 °C for derivatization. After cooling to room temperature, the solution was filtered through a 0.22 μm filter prior to further analysis.

The determination of GO and MGO in the samples was performed with a C18 column (Capcell PAK C18 AQ, 4.6 mm I.D. × 250 mm, Agilent, Santa Clara, CA, USA) on an HPLC system (Agilent 1260, Agilent Inc., Santa Clara, CA, USA) equipped with a diode array detector (DAD). Gradient elution was carried out using acetonitrile as mobile phase A and 0.15% acetic acid in water as mobile phase B. The elution procedure of mobile phase A was as follows: 0–10 min, 8–40%; 10–12 min, 40–48%; 12–13 min, 48–60%; 13–15.5 min, 60–80%; 15.5–20.5 min, 80–8%. The concentrations of GO and MGO were calculated based on a standard curve using 5-MQ as the internal standard.

### 2.5. Determination of Free and Protein-Bound CML and CEL

Free and protein-bound CML and CEL in fish samples were extracted following the reports of Sun et al. [24] and Niu et al. [25]. Approximately 1.00 g of the grounded fish sample was mixed with 10 mL of 10% trichloroacetic acid and spiked with internal standards (D_4_-CML and D_4_-CEL), then homogenized in an ice-water bath. After being centrifuged for 10 min at 2000× *g*, the collected supernatant was mixed with 10 mL hexane and then centrifuged to remove fat. The remaining mixture was loaded into a pre-activated MCX column (60 mg/3 mL, ANPEL Laboratory Technologies Inc., Shanghai, China), and the sample containing free CML and CEL was eluted with 5% ammonia in methanol. The elution was then blow-dried under nitrogen at 50 °C and reconstituted in 1 mL of 80% methanol. Since the heat treatment has no significant effect on free CML and CEL levels in fish [16], the content of free CML and CEL in prefabricated aquatic products after heating was not determined in this paper. 

The protein-bound CML and CEL in collected commercially prepared, as well as in raw and heat-treated prefabricated, fish products were extracted as follows. Around 0.21 g of the grounded sample was mixed with 2 mL of 0.2 mol/L borate buffer and 0.4 mL of 2.0 mol/L sodium borohydride solution then homogenized. The mixture was reduced for 8 h at 4 °C, and then 4 mL of chloroform–methanol solution (*v*:*v*, 2:1) was added, mixed, and centrifuged for 10 min at 2000× *g*. The precipitated protein was collected and hydrolyzed in 4 mL of 6 mol/L HCl for 24 h at 110 °C. The hydrolysate was spiked with internal standards and vacuum-dried at 50 °C then redissolved in water. After being cleaned with an MCX column, the sample was nitrogen-dried and reconstituted in 1 mL of 80% methanol.

Each of the final samples containing CML and CEL was cleaned with a 0.22 μm filter before analyzation with HPLC-MS/MS (liquid chromatography, A-30 Altus, PerkinElmer, Inc., Waltham, MA, USA; mass spectrometer, Qtrap 4500, AB Sciex Pte. Ltd., Singapore). In brief, a volume of 10 μL of the tested sample was injected into an HILIC column (Atlantis, Waters Corp, Milford, MA, USA), and the liquid chromatography separation conditions were employed as described in the study by Sun et al. [24]. Settings for the mass spectrometric analysis were based on the method reported by Wu et al. [26]. Quantification of CML and CEL in the samples was conducted based on the corresponding standard curve, utilizing the same levels of internal standards.

### 2.6. Statistic Analysis

Since the moisture levels varied significantly among different fish samples, the contents of GO, MGO, CML, and CEL were calculated based on the weight of dry matter (sample weight after removing moisture). The detection of GO, MGO, CML, and CEL was performed in duplicate, with each experiment repeated twice. All the data were reported as mean ± standard deviation. For Figure 1, Figure 2 and Figure 3, the standard deviation of data from the replicated experiments was used as an error bar. To analyze the significant differences (*p* < 0.05) among the data, a Duncan test was conducted using SPSS software (SPSS Statistics 22.0, IBM Corp., Armonk, NY, USA). Correlation analysis was performed using Origin software (OriginPro 2023b, OriginLab Corp., Northampton, MA, USA).

## 3. Results and Discussion

### 3.1. Chemical Composition of Commercially Prepared and Prefabricated Fish Products

The moisture, protein, fat, and salt levels of the 32 collected fish products are presented in Table 2 based on the wet weight of the sample. Moisture contents varied significantly among different samples, ranging from 6.53% ± 0.02% to 88.09% ± 0.04%. Prefabricated fish products exhibited relatively higher moisture levels (ranging from 30.08% to 88.09%, with an average content of 62.19% ± 16.72%) compared to commercially prepared products (ranging from 6.53% to 76.21%, with an average content of 31.84% ± 19.55%). Specifically, samples 1–4, which were manufactured through frying or roasting, displayed lower moisture levels compared to both commercially sterilized samples (5–13) and raw samples (14–34). The protein contents of the collected samples ranged from 10.41% to 57.15%, with an average value of 29.07% ± 12.65%. Similarly, the fat contents varied significantly, ranging from 0.6% to 39.65%, with an average value of 9.77% ± 9.70%. 

Salt contents in the collected samples ranged from 0.12% ± 0.02% to 11.52% ± 0.45%. Products 1–8 were categorized as fish food snacks with relatively high salt contents (1.39–9.13%). Canned fish products (10–13), except for sample 9 (2.96%), had generally lower salt contents (0.83–1.43%). Products stored at room temperature (14–23), regardless of the packaging form, had higher salt contents (4.91–11.52%). The vacuum-packaged products (24–34) for frozen storage exhibited reduced salt content ranging from 0.12% to 1.78%, demonstrating an extended shelf life. It is advisable to avoid the frequent consumption of snack products and salted fish products that are stored at room temperature to reduce excessive sodium intake.

### 3.2. Contents of GO and MGO in Commercially Prepared and Prefabricated Fish Products

GO and MGO are the two most representative α-dicarbonyl compounds found in food, commonly generated from the Maillard reaction, lipid oxidation, and degradation of reducing sugars during food processing [27,28]. Figure 1 shows the contents of GO and MGO in commercially prepared fish products as well as prefabricated fish products before and after heating. In the category of commercially prepared fish products, snack foods (samples 1–8) exhibited higher levels of GO and MGO, with GO ranging from 25.00 ± 3.34 to 137.12 ± 25.87 mg/kg of dry matter and MGO ranging from 11.47 ± 1.39 to 43.23 ± 7.91 mg/kg of dry matter. In canned fish products (samples 9–13), relatively lower levels of GO and MGO were observed, with GO ranging from 21.43 ± 2.50 to 43.10 ± 2.00 mg/kg of dry matter and MGO ranging from 8.12 ± 1.04 to 20.07 ± 0.51 mg/kg of dry matter. In the prefabricated fish products (samples 14–34), there were large differences in the levels of GO (ranging from 3.58 ± 0.85 to 72.81 ± 12.86 mg/kg of dry matter) and MGO (ranging from 3.05 ± 0.13 to 25.43 ± 0.98 mg/kg of dry matter) among different samples. Higher contents of GO and MGO in snack food (samples 1–8) may be related to their high salt levels (1.39–9.13%, Table 2). Meng et al. [29] and Li et al. [22] found no significant influence of salt (0.5–2.5% and 2%) on the changes of GO and MGO during the heating of grass carp and pork. However, Kocadağlı and Gökmen [30] reported that the addition of 1% NaCl increased the concentration of MGO and 1-deoxyglucose in biscuits compared to the control group, while other α-dicarbonyl compounds were not affected. 

The effects of heating on the levels of GO and MGO in prefabricated fish products were inconsistent. GO contents increased by 8.83–37.47% after heating in samples 18, 22, 26–28, and 31 but decreased by 7.80–38.56% in samples 15–17, 19, 20, 25, 29, 30, and 32. After heating, the MGO content increased by 9.65–71.76% in samples 14, 18, 22–28, 31, and 32 (*p* < 0.05) and decreased by 6.36–22.99% in samples 15, 17, 29, and 30 (Figure 1). Zhu et al. [31] suggested that frying and boiling increased the levels of GO and MGO in chicken meat, whereas sterilization reduced the MGO content. Li et al. [22] observed a reduction of 16% and 13% in the average amounts of GO and MGO, respectively, in sterile pork (121 °C, 10 min) compared to raw pork. As reactive intermediates of lipid peroxidation and sugar autoxidation [31], α-dicarbonyl compounds could rapidly combine with other substances, e.g., lysine, histidine, arginine, and cysteine residues, to form more stable compounds such as AGEs and heterocyclic amines [32]. This process is also facilitated by thermal treatment [27]. The variations in the amounts of GO and MGO are dependent on their formation and reaction rate, which are influenced by various factors such as temperature and the presence of amino acids, proteins, fats, etc. [33,34]. The changes in GO and MGO levels before and after heating may differ due to the use of different ingredients and materials in the prefabricated fish products examined in this study.

### 3.3. Contents of Free CML and CEL in Commercially Prepared and Prefabricated Fish Products

Similar to the findings of Niu et al. [3], significant variations were observed in the levels of free CML and CEL among different types of fish products (Figure 2). Free CML content ranged from 0.11 ± 0.02 to 3.29 ± 0.74 mg/kg of dry matter, and CEL ranged from 0.03 ± 0.00 to 2.19 ± 0.03 mg/kg of dry matter. Samples 9 and 11 of the commercially prepared fish products exhibited significantly higher levels of free CML and CEL compared to other samples. This may be attributed to the addition of fermented black beans in these two canned products. Fermented black beans are rich in protein and various amino acids [35], consequently increasing the levels of free CML and CEL in canned fish products. Samples 14, 18, 19, 23, 25, 26, and 32 of the prefabricated fish products had relatively high levels of free AGEs; however, no discernible pattern was observed, which may arise from the utilization of different raw materials and additives.

### 3.4. Contents of Protein-Bound CML and CEL in Commercially Prepared and Prefabricated Fish Products

The protein-bound CML and CEL levels in samples 1–13 of the commercially prepared fish products ranged from 16.69 ± 0.83 to 50.47 ± 7.49 mg/kg of dry matter and 45.01 ± 1.09 to 128.32 ± 20.33 mg/kg of dry matter, respectively (Figure 3). The data suggests that processed fish products typically exhibit elevated levels of AGEs, which may be attributed to their low moisture content as well as the impact of heating. Tavares et al. [36] also discovered a positive linear correlation between the content of AGEs and the moisture levels in hairtail fillets. The Maillard reaction is impeded under high moisture conditions, while lower moisture content facilitates the chemical reactions that contribute to the accumulation of AGEs during processing and storage [8].

The levels of protein-bound AGEs in samples 14–32 of the prefabricated fish products were comparatively lower than those in commercially prepared ones (samples 1–13), ranging from 1.20 ± 0.29 to 28.38 ± 0.42 mg/kg of dry matter for CML and from 2.64 ± 0.75 to 32.18 ± 0.99 mg/kg of dry matter for CEL. Among them, the frozen products (samples 20–32) generally had lower CML and CEL levels (CML: 1.20 ± 0.29–7.50 ± 1.52 mg/kg of dry matter; CEL: 2.64 ± 0.75–26.72 ± 7.78 mg/kg of dry matter) compared to products stored at room temperature (samples 14–19, CML: 9.53 ± 0.35–28.38 ± 0.42 mg/kg of dry matter; CEL: 10.73 ± 1.69–32.18 ± 0.99 mg/kg of dry matter) (Figure 3). Chemical changes like the Maillard reaction and protein and fat oxidation that account for the formation of AGEs could be accelerated by high temperatures, promoting an accumulation in AGEs during storage [4,37].

Heat treatment increased the levels of protein-bound CML and CEL, respectively, by 2.98-fold and 4.34-fold in prefabricated fish products, which is consistent with previous research findings [3,24]. Proteins involved in the Maillard reaction undergo thermal denaturation, resulting in the exposure of amino acid residues and ultimately leading to increased glycation rates and browning intensity [38]. Additionally, the oxidation of fat in meat is highly susceptible to elevated temperatures, which leads to the production of numerous aldehydes, ketones, and other intermediates that are ultimately involved in the formation of AGEs [39].

There’s increasing evidence indicating that the consumption of dietary AGEs can contribute to elevated levels of AGEs in human serum and urine [5,6,8]. Studies conducted on mice have also demonstrated that the intake of glycated proteins resulted in increased levels of AGEs in various organs including the heart, kidneys, liver, lungs, and spleen [40]. Although there are currently no established limits for AGE levels in food, it is possible that excessive exposure to high levels of dietary AGEs may pose certain health risks. Therefore, it is advisable to reduce the daily consumption of commercially prepared fish products, particularly those with high salt content or those subjected to extensive heating processes, in order to minimize an excessive intake of AGEs. 

### 3.5. Correlation Analysis between the Levels of GO, MGO, AGEs, and Chemical Composition in Commercially Prepared and Prefabricated Fish Products

Since AGEs can be formed through the Maillard reaction and lipid peroxidation, which are two reactions that are influenced by many factors such as food components, processing, and storage conditions [8,12], a correlation analysis was conducted to find the underlying connections between AGEs and their precursors (GO, MGO), as well as their relations with chemical compositions, especially salt, in the collected fish products. Prefabricated samples (14–32) were analyzed separately from commercially prepared ones to investigate the influence of thermal treatment on AGE formation. 

In commercially prepared fish products (Figure 4a), significant positive correlations were found in the levels of GO and MGO (R = 0.955, *p* < 0.05), free CML and CEL (R = 0.736, *p* < 0.05), as well as protein-bound CML and CEL (R = 0.934, *p* < 0.05), suggesting the similarity in the formation pathways of these compounds. Surprisingly, both GO and MGO demonstrated a negative correlation with fat contents (R = −0.720 and −0.751, *p* < 0.05). A negative relation between GO and fat levels was also been reported by Wang et al. [41] but no precise reason was given. As indicated in Table 1, the majority of commercially prepared samples were found to contain granulated sugar. Therefore, it can be inferred that the presence of GO and MGO in these samples primarily resulted from the degradation of sugars and Maillard reaction products upon thermal treatment, rather than lipid oxidation. Furthermore, the contents of GO and MGO displayed positive correlations with salt levels (R = 0.619 and 0.603, *p* < 0.05), further confirming the aforementioned finding that salt could increase the formation of α-dicarbonyl compounds by promoting glucose dehydration. No significant relationship was observed between the levels of GO/MGO and free or protein-bound CML/CEL (R = −0.152–0.154, *p* > 0.05).

In contrast to commercially prepared fish products, the levels of GO and MGO in raw prefabricated samples exhibited a positive correlation with the fat contents (R = 0.526 and 0.521, *p* < 0.05), respectively (Figure 4b). This could indicate that the formation of GO and MGO at low temperatures is likely predominantly driven by lipid oxidation and not Maillard reaction or sugar degradation. Similar to that found in the commercially prepared samples, there was no obvious correlation between the levels of GO/MGO and free or protein-bound CML/CEL in prefabricated ones before heating (R = −0.272–0.052, *p* > 0.05).

Conversely from the finding in commercially prepared products, salt suppressed the heat-induced variations of GO and MGO in the prefabricated ones, with correlation coefficients of −0.377 (*p* > 0.05) and −0.470 (*p* < 0.05), respectively (Figure 4b). However, when prefabricated samples containing granulated sugar were analyzed separately, the changes in GO and MGO upon heating were positively correlated with salt levels (R = 0.043 and 0.242); however, there was no statistical difference. According to Liang et al. [42], in the presence of Na+, D-glucose underwent 1,2-dehydration to form α-dicarbonyl compounds. In this regard, we assumed that the impact of NaCl on the formation of α-dicarbonyl compounds was related to the presence of sugars. Further study should be conducted to confirm this hypothesis. Additionally, positive correlations were found between salt levels and the heat-induced formation of CML and CEL (R = 0.333 and 0.202) in prefabricated samples (Figure 4b), although the difference was not statistically significant (*p* > 0.05). This result aligned with the study of Niu et al. [16], which found that the addition of 1.5–5% NaCl led to an increase in the formation of protein-bound CML and CEL in sterilized pork samples. Similarly, NaCl has long been recognized as a pro-oxidant in meat products due to its ability to release iron ions from heme proteins and catalyze the oxidation of fat and protein [14]. 

The heat-induced (100 °C, 15 min) changes in the contents of GO and MGO in prefabricated fish products were positively correlated with each other (R = 0.674, *p* < 0.05, Figure 4b), indicating that both compounds were similarly influenced by the heating process in terms of their formation. Variations in GO and MGO contents after heating exhibited a negative correlation with the heat-induced formation of protein-bound CML and CEL (R = −0.289–−0.558), even though the correlations were not statistically significant, except for the correlation between GO and CML. This suggested that GO and MGO were likely to be involved in the generation of protein-bound CML and CEL. Moreover, a negative correlation was observed between protein levels in prefabricated fish products and heat-induced variations in GO and MGO, and a positive correlation was found with the heat-induced formation of CML and CEL. This implied that proteins may contribute reactive amino groups for reactions with GO and MGO, leading to the generation of protein-bound CML and CEL.

## 4. Conclusions

Large variations were observed in the levels of α-dicarbonyl compounds (GO, MGO) and free and protein-bound AGEs (CML, CEL) in commercially prepared fish products and prefabricated ones. Snack food with high salt content exhibited higher levels of GO and MGO compared to canned fish products and prefabricated ones. Variations in the contents of free CML and CEL in the collected samples increased 29.9- and 73.0-fold. In addition, higher levels of protein-bound CML and CEL were found in commercially prepared products, attributed to the thermal treatment of these samples. Correlation analysis revealed a negative relationship between GO/MGO and fat levels in commercially prepared samples, and a positive relationship was found in raw prefabricated ones. These findings suggested that the formation pathway of GO and MGO in fish products was likely influenced by the processing and storage temperature rather than fat levels. The generation of protein-bound CML and CEL during heating was mainly related to the reactions of GO and MGO with reactive amino groups in proteins, as indicated by their negative correlation with heat-induced changes of GO and MGO, as well as the positive correlation with protein levels. The impact of NaCl on α-dicarbonyl compound formation may be correlated to the presence of sugars; however, this requires further confirmation.

## Figures and Tables

**Figure 1 foods-12-04324-f001:**
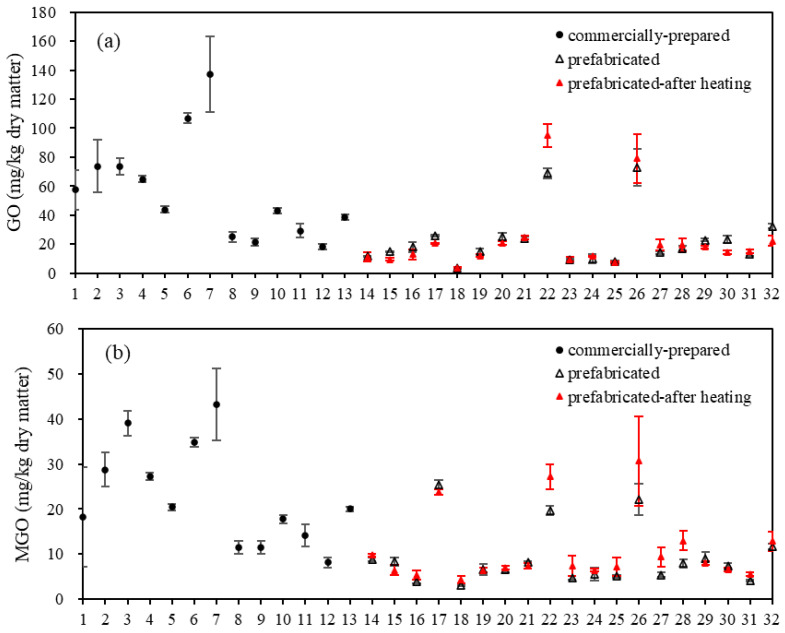
Levels of (**a**) glyoxal (GO) and (**b**) methylglyoxal (MGO) in commercially prepared and raw and heat-treated (100 °C, 15 min) prefabricated fish products (n = 2).

**Figure 2 foods-12-04324-f002:**
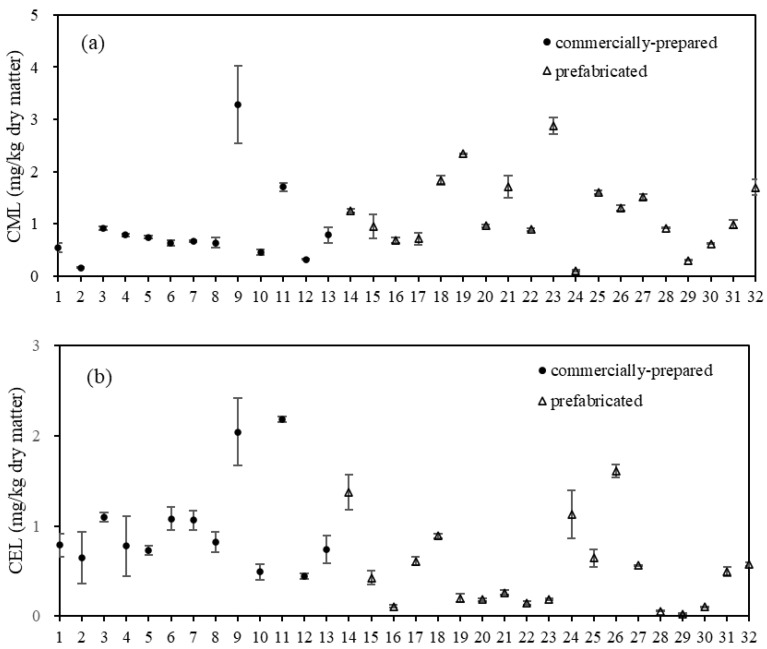
Levels of free (**a**) N^ε^-carboxymethyllysine (CML) and (**b**) N^ε^-carboxyethyllysine (CEL) in commercially prepared and raw prefabricated fish products (n = 2).

**Figure 3 foods-12-04324-f003:**
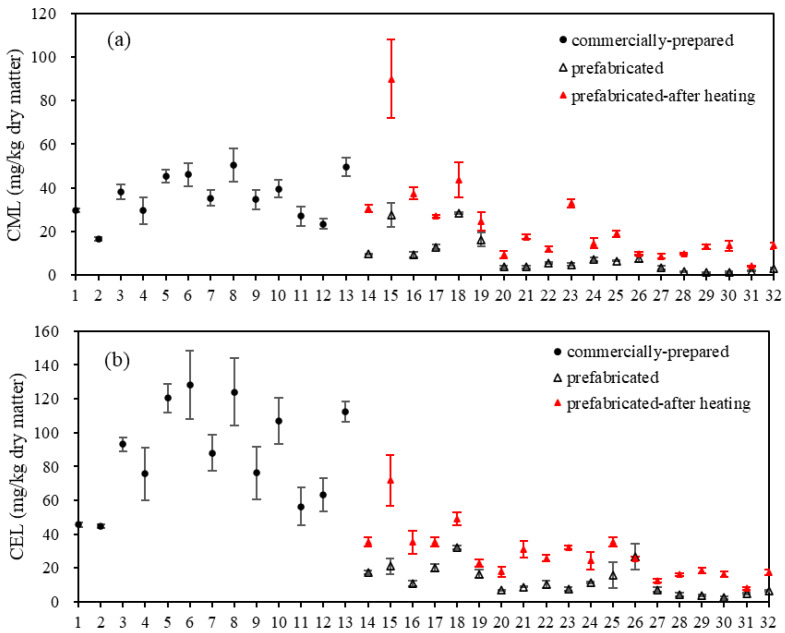
Levels of protein-bound (**a**) N^ε^-carboxymethyllysine (CML) and (**b**) N^ε^-carboxyethyllysine (CEL) in commercially prepared and raw and heat-treated (100 °C, 15 min) prefabricated fish products (n = 2).

**Figure 4 foods-12-04324-f004:**
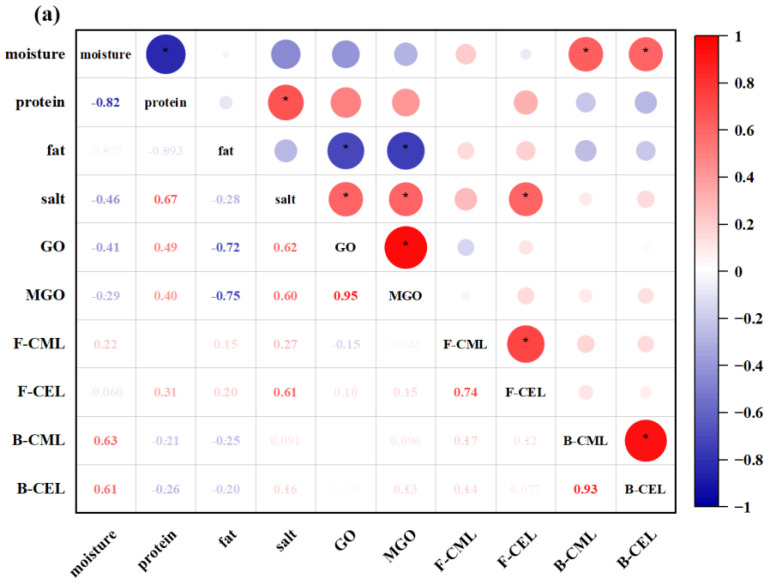
Correlation between chemical composition and glyoxal (GO), methylglyoxal (MGO), free/protein-bound N^ε^-carboxymethyl-lysine (CML), and N^ε^-carboxyethyl-lysine (CEL) levels in (**a**) commercially prepared fish products and (**b**) prefabricated fish products before and after heating (100 °C, 15 min). F-CML and F-CEL refer to free CML and CEL, respectively; B-CML and B-CEL refer to protein-bound CML and CEL, respectively; H-GO, H-MGO, H-CML, and H-CEL refer to heat-induced variations of GO, MGO, CML, and CEL, respectively. * denotes a significant difference (*p* < 0.05) between the correlated data. Different size of the circles in the figure represents different magnitude of the correlation value.

**Table 1 foods-12-04324-t001:** Information of 32 collected commercial fish products from local markets in Shanghai, China.

Sample No.	Storage Condition	Package	Heating Method ^1^	Shelf Life (Month)	Raw Material	Ingredients
1	RT	Regular	Frying	9	Yellow croaker	Salt, oil, white granulated sugar, starch, MSG
2	RT	Regular	Roasting	12	*Liza haematocheila*	Salt, white granulated sugar, trehalose
3	RT	Vacuum	Roasting	12	Yellow croaker	Salt, white granulated sugar, MSG, spices
4	RT	Regular	Roasting	9	Yellow croaker	Salt, white granulated sugar, MSG, spices
5	RT	Vacuum	CS	9	Yellow croaker	Salt, oil, white granulated sugar, soy sauce, MSG, starch, spices
6	RT	Vacuum	CS	7	Yellow trevally	Salt, oil, pepper, spices
7	RT	Vacuum	CS	12	Yellow trevally	Salt, oil, white granulated sugar, pepper, sesame, spices
8	RT	Vacuum	CS	12	Silver carp	Salt, oil, white granulated sugar, pepper, fermented beans, sesame, spices
9	RT	Canned	CS	24	Yellow croaker	Salt, oil, white granulated sugar, fermented beans, soy sauce, rice wine, spices
10	RT	Canned	CS	36	Sardine	Salt, water, tomato juice, xanthan gum
11	RT	Canned	CS	36	Dace	Salt, oil, white granulated sugar, fermented beans, soy sauce, water, spices
12	RT	Canned	CS	24	Anchovy	Salt, oil, white granulated sugar, soy sauce, rice wine, spices
13	RT	Canned	CS	36	Tuna	Salt, water, vegetable juice, MSG
14	RT	Vacuum	Raw	6	Sea eel	Salt
15	RT	Unpackaged	Raw	1	Yellow croaker	Salt
16	RT	Vacuum	Raw	6	Black carp	Salt, Baijiu, MSG, spices
17	RT	Unpackaged	Raw	1	*Stolephorus commersonnii*	Salt
18	RT	Vacuum	Raw	6	Yellow croaker	Salt
19	RT	Vacuum	Raw	5	Black carp	Salt, Baijiu, spices
20	Frozen	Unpackaged	Raw	--	Black carp	Salt
21	Frozen	Unpackaged	Raw	--	Black carp	Salt
22	Frozen	Vacuum	Raw	24	Sea eel	White granulated sugar, soy sauce, water, glucose syrup, fructose syrup, edible alcohol
23	Frozen	Vacuum	Raw	12	Basa fish	Salt, water
24	Frozen	Vacuum	Raw	12	Snakehead	Salt, white granulated sugar, starch
25	Frozen	Vacuum	Raw	12	Snakehead	Salt, trehalose
26	Frozen	Vacuum	Raw	24	Sea eel	White granulated sugar, soy sauce, glucose syrup, fructose syrup, edible alcohol
27	Frozen	Vacuum	Raw	12	Culter alburnus	Salt, cooking wine, MSG, spices
28	Frozen	Vacuum	Raw	18	Yellow croaker	Salt
29	Frozen	Vacuum	Raw	18	Yellow croaker	Salt, white granulated sugar, MSG
30	Frozen	Vacuum	Raw	18	Yellow croaker	Salt, white granulated sugar, MSG
31	Frozen	Vacuum	Raw	12	Whitefish	Salt, spices
32	Frozen	Vacuum	Raw	18	Bass	Salt, Baijiu, spices

^1^ CS: Commercial sterilization.

**Table 2 foods-12-04324-t002:** Contents of moisture, protein, fat, and salt in salted commercial fish products (n = 3).

Sample No.	Moisture (%)	Protein (%)	Fat (%)	Salt (%)
1	6.53 ± 0.02	57.15 ± 1.40	17.10 ± 0.00	2.59 ± 0.17
2	15.92 ± 0.21	44.81 ± 0.08	1.45 ± 0.05	2.91 ± 0.25
3	18.51 ± 0.07	45.51 ± 0.82	10.15 ± 0.05	3.36 ± 0.09
4	12.89 ± 0.07	53.86 ± 0.68	8.45 ± 0.05	3.40 ± 0.37
5	43.53 ± 0.40	28.71 ± 0.03	14.35 ± 0.15	1.39 ± 0.07
6	23.98 ± 0.04	37.11 ± 0.05	6.35 ± 0.05	9.13 ± 0.22
7	25.71 ± 0.05	31.19 ± 1.22	6.30 ± 0.10	8.97 ± 0.70
8	34.67 ± 0.20	34.16 ± 0.05	21.15 ± 0.05	2.96 ± 0.11
9	31.21 ± 0.77	32.01 ± 0.35	27.40 ± 0.00	2.96 ± 0.44
10	67.30 ± 0.65	19.35 ± 0.24	9.50 ± 0.10	0.83 ± 0.12
11	33.70 ± 4.19	25.50 ± 1.53	35.25 ± 0.05	1.43 ± 0.28
12	23.74 ± 0.32	24.65 ± 0.07	39.65 ± 0.15	0.88 ± 0.03
13	76.21 ± 0.04	23.48 ± 0.05	1.05 ± 0.05	1.35 ± 0.25
14	51.43 ± 0.31	28.85 ± 0.16	4.65 ± 0.05	11.52 ± 0.45
15	48.94 ± 0.23	31.46 ± 0.66	4.35 ± 0.05	5.41 ± 0.57
16	51.56 ± 0.26	30.19 ± 0.19	3.50 ± 0.00	6.19 ± 0.25
17	39.44 ± 0.40	44.09 ± 0.72	6.65 ± 0.05	4.83 ± 0.09
18	58.70 ± 0.44	29.91 ± 0.08	1.95 ± 0.05	4.83 ± 0.09
19	48.21 ± 0.36	32.41 ± 0.08	8.10 ± 0.00	5.36 ± 0.28
20	62.64 ± 1.04	25.78 ± 0.17	4.30 ± 0.00	8.78 ± 0.26
21	70.56 ± 0.00	23.50 ± 0.12	3.90 ± 0.10	8.41 ± 0.03
22	58.47 ± 0.59	20.18 ± 0.41	18.25 ± 0.15	1.01 ± 0.02
23	88.09 ± 0.04	12.66 ± 0.11	0.60 ± 0.00	0.73 ± 0.14
24	84.58 ± 0.26	10.41 ± 0.03	3.15 ± 0.05	1.14 ± 0.06
25	84.14 ± 0.48	10.52 ± 1.32	3.45 ± 0.05	0.63 ± 0.04
26	51.06 ± 0.63	16.12 ± 0.26	22.70 ± 0.10	1.21 ± 0.06
27	80.77 ± 0.04	17.56 ± 0.18	0.80 ± 0.00	1.39 ± 0.19
28	72.18 ± 0.07	18.97 ± 0.35	10.00 ± 0.00	0.13 ± 0.02
29	74.30 ± 0.17	19.27 ± 0.15	11.40 ± 0.00	0.12 ± 0.02
30	65.33 ± 1.44	18.53 ± 0.21	12.50 ± 0.00	1.54 ± 0.06
31	72.87 ± 0.78	18.43 ± 0.03	11.15 ± 0.05	1.51 ± 0.06
32	79.45 ± 0.01	17.69 ± 0.36	0.90 ± 0.00	1.78 ± 0.03

## Data Availability

The data used to support the findings of this study can be made available by the corresponding author upon request.

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
