# Peer review of "Investigation of Advanced Glycation End-Products, α-Dicarbonyl Compounds, and Their Correlations with Chemical Composition and Salt Levels in Commercial Fish Products"

_foods, 2023, doi:10.3390/foods12234324_

Round 1

Reviewer 1 Report

Comments and Suggestions for Authors

Authors made the profile of AGEs in fish products and evaluated the effect of salt on the production of these compounds. It seems that the result is meaningful from the viewpoints of risk-assessment for human health. For publish, I'd like to ask following points and suggest to revise to authors.

・Firstly, authors showed the profile of AGEs compounds in various fish products. From these data, we can comprehensively understand the distribution of these compounds in each product. However, the impact of each concentration on human health has not been discussed. Why don't you put more information in discussion from the viewpoints of risk assessment against human health with referring some previous reports.

・In each analytical data, it is better to put the information of replicate for analysis in the section of Materials and Methods and/or legends in each Figure.

L118

・Here, the unit for time is written as minutes. But in other sentences, it is written as "min". It is better to unify.

Author Response

Thank you very much for taking the time to review this manuscript. The detailed responses can be found below and the corresponding revisions are highlighted in the re-submitted files.

Comments 1: Authors made the profile of AGEs in fish products and evaluated the effect of salt on the production of these compounds. It seems that the result is meaningful from the viewpoints of risk-assessment for human health. For publish, I'd like to ask following points and suggest to revise to authors. 

Response 1: Thank you for the recognition. 

Comments 2: Firstly, authors showed the profile of AGEs compounds in various fish products. From these data, we can comprehensively understand the distribution of these compounds in each product. However, the impact of each concentration on human health has not been discussed. Why don't you put more information in discussion from the viewpoints of risk assessment against human health with referring some previous reports. 

Response 2: So far, there are currently no established regulatory limits for levels of AGEs in food; however, mounting evidence suggests that exposure to elevated dietary AGEs may pose certain health risks. The consumption of dietary AGEs can contribute to increased levels of these compounds in human serum and urine. Studies conducted on mice have also demonstrated that the ingestion of glycated proteins leads to heightened concentrations of AGEs in vital organs such as the heart, kidneys, liver, lungs, and spleen. Therefore, it is imperative to reduce daily intake of commercially-prepared fish products—particularly those with high salt content or subjected to excessive heat treatment—in order to mitigate excessive uptake of AGEs. The aforementioned discussion was included in section 3.4, Line 297–305.

Comments 3: In each analytical data, it is better to put the information of replicate for analysis in the section of Materials and Methods and/or legends in each Figure.

Response 3: The replicate numbers for the corresponding analysis were provided in the legends of Figure 1–3, respectively.

Comments 4: L118
Here, the unit for time is written as minutes. But in other sentences, it is written as "min". It is better to unify.

Response 4: According to the reviewer’s comments, the uses of “minutes” have been changed to “min”, “hours” has been changed to “h”, respectively in Line 125, 134 and 124. 

Reviewer 2 Report

Comments and Suggestions for Authors

The reviewer’s remarks are highlighted in yellow in the attached file

 Introduction is well presented, but:

 Line 87:  not well explained why the authors emphasize the influence of salt? let them justify - random selection or expected influence??? are there other studies related to the influence of salt on the formation of these endogenous contaminants or not?

they are nowhere mentioned in the introduction. if any, let them be indicated/quoted.

 Materials and Methods

2.2. Sample preparation

Table 1 (line 111): it is not clear what is hidden under the name “Material”

please clarify what do you mean - type of fish, trade name of the analyzed product or other?

 2.3. Basic composition

Line 113: It is better to write chemical/proximate composition instead of basic composition

or macronutrients? please, rewrite subtitle and maybe Titles??  

 Results

Line 201: the data presented in the Table 2 for dry or wet weight are presented? to be written in the text or in the title of the Table 2

the presented data in each row in the Table do not add up to 100%, but are very different?

How then were these values determined - per 100 g of tissue, sample, something else?

Let it be explained!

 Conclusions

Line 366: correlation analysis determines the correlations, i.e. shows an increasing or decreasing trend, while association gives information about the influence of a particular variable

The two terms are different!

The conclusions are well written, please clarify the terms.

Author Response

Thank you very much for taking the time to review this manuscript. Please find the detailed responses below and the corresponding revisions highlighted in the re-submitted files.

Comments 1: Introduction is well presented, but:
Line 87:  not well explained why the authors emphasize the influence of salt? let them justify - random selection or expected influence??? are there other studies related to the influence of salt on the formation of these endogenous contaminants or not?
they are nowhere mentioned in the introduction. if any, let them be indicated/quoted.

Response 1: Thank you for pointing this out. The studies cited in the introduction section, lines 83–91, have examined the influence of salt on the formation of AGEs in meat products. Based on these findings, it is evident that the impact of salt on AGEs formation is intricate and requires further comprehensive investigation.

Comments 2: Materials and Methods
2.2. Sample preparation
Table 1 (line 111): it is not clear what is hidden under the name “Material”
please clarify what do you mean - type of fish, trade name of the analyzed product or other? 

Response 2: The term "Material" in Table 1 refers to the raw fish used in the products. To provide further clarification, it has been revised to "Raw material" .

Comments 3: 2.3. Basic composition
Line 113: It is better to write chemical/proximate composition instead of basic composition or macronutrients? please, rewrite subtitle and maybe Titles??

Response 3: “basic composition” have been modified as “chemical composition” in the title (Line 3) and subtitles (Line 120, 187) in the context. 

Comments 4: Results
Line 201: the data presented in the Table 2 for dry or wet weight are presented? to be written in the text or in the title of the Table 2

Response 4:  The data presented in Table 2 were based on the wet weight of the tested sample. It was illustrated in Line 189.

Comments 5: the presented data in each row in the Table do not add up to 100%, but are very different?
How then were these values determined - per 100 g of tissue, sample, something else?
Let it be explained!

Response 5: The data presented in Table 2 were calculated based on the weight of the tested sample, specifically representing grams per 100 grams of sample. Given that fish products consist of complex chemical substances such as moisture, protein, fat, salt, carbohydrates, minerals, vitamins, etc., it is reasonable to observe that the measured contents of moisture, protein, fat and salt in this study do not sum up to 100%.

Comments 6: Conclusions
Line 366: correlation analysis determines the correlations, i.e. shows an increasing or decreasing trend, while association gives information about the influence of a particular variable
The two terms are different!
The conclusions are well written, please clarify the terms.

Response 6: Thank you for pointing it out. The misusing of the term "association" was respectively modified in the Abstract, Line 25, 26; section 3.5, Line 319, 353, 371; Conclusions, Line 383, 384, 390.